# The impact of social support on career decision-making difficulties: The serial mediating roles of career decision-making self-efficacy and job search clarity, and moderation by proactive personality

**Rong Chen[1], Qin Zhang[1], Yunfei Cao** [2]*

**1** Mental Health Education Center, Chengdu Aeronautic Polytechnic University, Chengdu, China, **2** School of Teacher Education, Chengdu University, Chengdu, China

\* caoyunfei@cdu.edu.cn

## Abstract

Career decision-making difficulties are a common challenge for college students, which can hinder their transition from education to employment. Based on the Social Cognitive Model of Career Self-Management, this study explored how social support was related to career decision-making difficulties. A total of 991 vocational college students participated in this cross-sectional quantitative study. Social support was measured through the Perceived Social Support Scale, career decision-making self-efficacy with the Career Decision-Making Self-Efficacy Questionnaire, job search clarity via the Job Search Clarity Questionnaire, career decision-making difficulties using the Career Decision-Making Difficulties Questionnaire, and proactive personality by the Proactive Personality Scale. The results showed that: Job search clarity partially mediates the relationship between social support and career decision-making difficulties. The indirect effect was significant, $\beta = -0.08$, $p < 0.001$, accounted for 20% of the total effect. In addition, career decision-making self-efficacy and job search clarity jointly serve as serial mediators in the link between social support and career decision-making difficulties. The serial indirect effect was significant, $\beta = -0.12$, $p < 0.001$, accounting for 30% of the total effect, indicating that higher social support sequentially enhances career decision-making self-efficacy and job search clarity, which in turn reduces career decision-making difficulties. Furthermore, proactive personality moderates the mediating role of job search clarity, particularly in the second stage of the mediation. Specifically, the effect of job search clarity on career decision-making difficulties became more pronounced as proactive personality increased ($\beta = -0.04$, $p < 0.001$). These findings extend the social cognitive model of career self-management by demonstrating the sequential mediating roles of career decision-making self-efficacy and job search clarity, and the moderating role of proactive personality, in the link between social support and reduced career

**Data availability statement:** All relevant data are within the manuscript and its Supporting Information files.

**Funding:** This research was funded by the Sichuan Provincial Key Research Base for Philosophy and Social Sciences - Education Development Research Center at China West Normal University, grant number CJF21045.

**Competing interests:** The authors have declared that no competing interests exist.

decision-making difficulties. Practically, they highlight the importance of designing career interventions for vocational college students that simultaneously strengthen social support networks, enhance self-efficacy, and build job search clarity, with particular attention to students with lower proactive personality who may benefit most from such clarity-building activities.

## 1. Introduction

In recent years, university graduates in China are facing severe employment pressure [1]. While the employment rate among higher vocational college students in China has been consistently rising, as reported in the Chinese 3-Year Vocational College Graduates' Employment Annual Report (2024), a marked difference remains in employment quality [2]. For instance, higher vocational college students often exhibit lower levels of initial employment stability, job satisfaction, and congruence between their majors and jobs, compared to their undergraduate students [3].

This difference may due to the critical role of initial career decisions play in students' career development. Decisions made during this period significantly impact long-term career paths, job satisfaction, and sustainable career development [4]. However, global economic instability and labor market unpredictability significantly increase the complexities in students' career decision-making processes. These challenges ultimately adversely affect job quality and satisfaction [5]. Therefore, investigating the influencing factors and underlying mechanisms of Career decision-making difficulties (CDMD) among college students informs interventions for university-to-work transitions and advances self-determination theory in career contexts.

CDMD refer to cognitive and emotional barriers that impede individuals' ability to make effective career choices [6,7]. Gati et al. (1996) categorized these difficulties into three categories: lack of readiness, lack of information, and inconsistent information [8]. CDMD may cause individuals to decision-making avoidance and increase the risk of incongruent selections [6]. Among college students, CDMD correlates with job search onset [9]and reduces effort during career exploration activities [10]. Further, these difficulties impair long-term career adaptation, diminishing job involvement and career satisfaction [11]. Given the significant impact of CDMD on employment outcomes, social support emerges as a critical mitigating factor. Specifically, perceived support attenuates the link between proactive dispositions and CDMD severity in moderated mediation pathways [12].

### Social cognitive model of career self-management

The Social Cognitive Model of Career Self-Management (CSM) provides a comprehensive theoretical framework that explains the complex relationship between social support and CDMD [13]. Specifically, the CSM model conceptualizes that adaptive career behavior is formed from the interplay among environmental variables, individual variables, and self-regulation processes. The self-regulation components include self-efficacy, outcome expectations, and goal setting; environmental variables encompass social support and barriers; and individual variables primarily refer

to personality traits. The interplay of these factors significantly impacts how individuals explore and choose their career paths, often through a complex and dynamic process [14]. Based on CSM, the present study aims to explore the relationship between social support and CDMD among vocational college students. Furthermore, it examines the mediating effects of career decision-making self-efficacy (CDSE) and job-seeking goal clarity, as well as the moderating effect of proactive personality.

## Social support and career decision-making difficulties

Social support, encompassing psychological or material assistance gleaned from social relationships, plays a pivotal role in shaping individual well-being and fostering adaptive occupational behaviors [15]. For youth, parents, peers, and schools are the main sources of social support [11]. A large amount of research has confirmed the positive relationship between social support and individual adaptive occupational behavior [16–18]. Notably, young adults who receive a lot of support during their major selection process exhibit greater persistence in their chosen fields of study, underscoring the pivotal role of social support in critical academic decisions [11]. Furthermore, parental support has been shown to strongly predict positive expectations about career outcomes, subsequently fostering career exploration intentions among high school juniors and seniors [19].

Conversely, faced with negative workplace behaviors like career decision-making challenges, social support can serve as a protective factor [20]. Empirical evidence shows a negative correlation relationship between social support and CDMD. Individuals with higher levels of perceived social support often reports lower levels of CDMD [21]. Nota et al. (2007) demonstrated that social support attenuates CDMD, particularly highlighting that adolescents with higher levels of parental support experience fewer challenges in career decision-making [22]. This finding is supported by local studies, which indicate that both parental support and support from diverse sources can substantially alleviate CDMD among vocational college students. The more extensive the support from families, teachers, and peers, the lower the level of CDMD reported [23].

Based on the existing literature, it is evident that social support plays a significant role in influencing CDMD. However, the specific mechanisms and potential moderating factors involved in this relationship remain unclear. Therefore, we firstly hypothesize that social support will act as a mitigating factor, reducing the degree of CDMD among vocational college students.

## The mediating role of career decision-making self-efficacy

Self-efficacy fuels personal self-management, driving proactive career agency. It shapes career decisions while bridging personal traits and environmental pressures – a key mediator in the intelligent career framework [13]. When navigating career choices, individuals draw on self-efficacy – their confidence to plan and decide despite uncertainty [24]. CDSE plays a dual role in promoting adaptive career behaviors. By tapping into their skill set, they lower career anxiety and lock onto career paths – a process enabled by self-efficacy [25]. It also armors them against career-choice hurdles, turning obstacles into navigable terrain [5].

Yet, the findings regarding the mediating role of CDSE in the relationship between social support and CDMD are not entirely consistent. Researchers like Jemini-Gashi et al. (2021) [26] and Nota et al. (2007) [22] revealed that CDSE significantly mediated the relationship between social support and CDMD. On the other hand, some studies, such as one involving college students in Taiwan [27], failed to identify significant mediation effects. These inconsistent findings may originate from variations in sample characteristics, measurement tools, and research designs. For instance, Jemini-Gashi et al.'s study focused on a specific population of students. In contrast, the Taiwan study employed different measures for CDSE and CDMD.

Considering these inconsistent findings, this study aims to further explore the mediating role of CDSE in the relationship between social support and CDMD among vocational college students.

 

## The mediating role of job search clarity

Goals, as the driving force for intentional actions, have a profound impact on behavior, especially when they are clear, specific, and both challenging and attainable [28]. The CSM proposes that goals are vital elements of the career choice stage. They affect career decisions through the intricate interplay of self-efficacy, outcome expectations, goals, environmental support, and barriers [13]. Within this model, goals operate via multiple routes: a) directly impacting career decision-making actions and outcomes; b) acting as intermediaries through which social support affects decision outcomes; and c) forming a chain where social support affects self-efficacy, self-efficacy affects goals, and goals subsequently shape career choice outcomes. Despite extensive research on the relationship between CDSE and CDMD, few studies have investigated other social cognitive variables operating within the CDSE framework, such as job search goals, and their impact on career exploration actions and outcomes [29].

Job search clarity, a key predictor of successful employment, is defined by individuals having a clear understanding of their desired career goals, the types of occupations they aspire to, and a clear vision of the work they wish to undertake [30]. Individuals with high job search goal clarity invest more time and effort in the career selection process, leading to quicker success in employment [31,32]. On the contrary, individuals lacking clear job search goals face difficulties in making career decisions due to insufficient knowledge about their career interests, goals, and job-related information, resulting in heightened challenges in career decision-making [33]. Empirical evidence indicates a significant, moderately negative correlation between job search clarity and CDMD among college students. Higher job search clarity scores were associated with lower levels of CDMD, underlining the importance of goal clarity in navigating career decisions [33]. Furthermore, self-efficacy displays a significant and positive correlation with job search clarity among college students. Increased career self-efficacy is related to increased job search clarity, indicating that as confidence in career-related abilities rises, students gain a clearer understanding of their job search goals [34]. To further explore the mechanisms underlying CDMD, this study examines the mediating role of job search clarity in the relationship between social support and CDMD.

## The moderating effect of proactive personality

Individual behavioral trends and tendencies act as a moderator in the transition from personal competence beliefs to actual job-search behavior [35]. The CSM suggests that personality factors are crucial in shaping adaptive occupational behavior, either by promoting or impeding behavioral performance [13]. Therefore, personality factors are likely to influence the process factors that lead to occupational behavioral outcomes. In recent years, proactive traits have garnered substantial interest from researchers in occupational psychology. Proactive personality, which includes traits where individuals take the initiative and influence their environment, has been recognized as a driving force for environmental change [36].

Some researchers have examined the moderating role of proactive personality in the relationship between job-search self-efficacy, job-search clarity, and proactive job-search behavior. Their findings show that when proactive personality levels are high, both job-search clarity and job-search self-efficacy have significant impacts on proactive job-search behavior. On the contrary, when proactive personality levels are low, the influence of job-search clarity and job-search self-efficacy on proactive job-search behavior decreases [37].

Surprisingly, there is a lack of studies exploring the moderating role of proactive personality in the relationship between CDSE, job-search clarity, and CDMD. A single study that touched on a similar topic looked at the moderating role of the Big Five personality traits in the relationship between CDSE and CDMD among college students [38]. The results showed that conscientiousness, one of the Big Five traits, moderated the relationship between college students' CDSE and CDMD. Specifically, when students had higher levels of both self-efficacy and responsibility, they felt less discomfort in making career decisions.

Proactive personality, closely related to openness, conscientiousness, extraversion, and agreeableness in the Big Five personality traits [39], is seen as an extension and deepening of the meanings of the Big Five personality model [40]. Given the importance of proactive personality in influencing adaptive career behaviors, this study aims to explore its moderating role in the relationship between CDSE, job-search clarity, and CDMD among vocational college students.

The current study aims to provide a comprehensive understanding of the processes underlying career decision-making behavior among vocational college students. Based on the social cognitive model of CSM, this study combines environmental variables, personality variables, and social cognitive variables. It systematically integrates social support, CDSE, job-search clarity, and proactive personality to clarify the processes behind the career decision-making behavior of vocational college students. The theoretical model of the study is shown in Fig 1 and hypotheses are as follows:

**Hypothesis 1**: Social support is negatively related to CDMD among vocational college students.

**Hypothesis 2a**: CDSE mediates the relationship between social support and vocational college students' CDMD.

**Hypothesis 2b**: Job search clarity mediates the relationship between social support and CDMD among vocational college students.

**Hypothesis 2c**: CDSE and job search clarity play a serial mediating role in the relationship between social support and vocational college students' CDMD.

**Hypothesis 3a**: Proactive personality moderates the mediating role of CDSE in the relationship between social support and CDMD among vocational college students. Specifically, the indirect effect via CDSE is stronger when proactive personality is higher.

**Hypothesis 3b**: Proactive personality moderates the mediating role of job search clarity in the relationship between social support and CDMD among vocational college students. Specifically, the mediating effect is stronger when proactive personality is higher.

## 2. Materials and methods

### 2.1. Participants and procedure

A cross-sectional survey was conducted to examine the relationships among social support, career decision-making self-efficacy (CDSE), job search clarity, proactive personality, and career decision-making difficulties (CDMD) among vocational college students. The study was conducted at a vocational university in Chengdu, China, from November 1 to November 30, 2023. Using a convenience sampling method, all students enrolled in the university during the study period were invited to participate in a survey through an online platform. Participants were eligible if they met the following

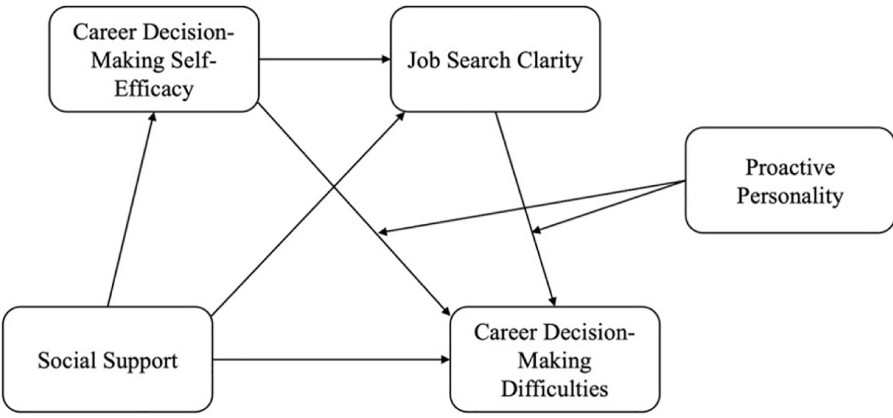

**Fig 1. Theoretical model.**

criteria: (a) full-time vocational college students, (b) currently enrolled in academic programs, (c) willing to provide informed consent, and (d) aged 18 years or older. Exclusion criteria included part-time students and those who had already secured full-time employment upon graduation. The sample size was determined based on prior studies examining structural equation modeling in career-related research, with a recommended minimum of 10–20 participants per estimated parameter. A sample size of 800 would provide adequate statistical power (≥.80) to detect medium-sized mediation and moderation effect. A total of 991 valid questionnaires were gathered, with an effective response rate of 84.56%.

The study was approved by the ethics committee of the School of Teacher Education at Chengdu University. The research procedures were in strict compliance with the ethical guidelines set forth in the Declaration of Helsinki (1989) and its later amendments, so as to ensure the protection of participants' rights and well-being. In an online survey, all participants were informed about the study's purpose, methods, content, and duration, and all study participants provided their informed consent by clicking a button. Data collection was coordinated with faculty members who were briefed on ethical protocols, though no formal training was required given the automated nature of the online survey.

### 2.2. Measures

**2.2.1. Career decision-making difficulties questionnaire.** The Chinese version of the Career Decision-making Difficulties Questionnaire (CDDQ, originally developed by Gati et al., 1996) [8], revised by Xue-Ping Shen in 2005 [41], was used for self-reporting. Respondents rated items on a 9-point Likert-type scale ranging from 1 (does not describe me at all) to 9 (describes me well). The questionnaire, which includes 35 items (3 validity-test items), assesses difficulties across three dimensions: lack of readiness, lack of information, and inconsistent information. The internal consistency of the three subscales in the revised Chinese version of the CDDQ was strong, with reliability coefficients ranging from 0.825 to 0.859. The overall scale's internal consistency was 0.866. In this study, the internal consistency of the three subscales was 0.65, 0.96, and 0.94, respectively. The entire scale had a high reliability coefficient of 0.94.

**2.2.2. The perceived social support scale.** The Perceived Social Support Scale(PSSS, originally developed by Zimet et al., 1988) [42], revised by Jiang Qianjin in 1999 [43], was employed. To better suit the participants, the terms "leaders, relatives, and colleagues" were adapted to "teachers, relatives, and classmates." The scale includes three dimensions: family support, friend support, and other support (teachers, classmates, and relatives), with a total of 12 items. Items were rated on a 7-point Likert-type scale from 1 (strongly disagree) to 7 (strongly agree). Higher scores indicate higher levels of perceived social support. The internal consistency coefficients for the three subscales were 0.82, 0.85, and 0.83, respectively. The overall scale had a high internal consistency coefficient of 0.92.

**2.2.3. The career decision-making self-efficacy scale.** The "Career Decision-making Self-Efficacy Questionnaire for College Students (CDSE, originally developed by Taylor et al., 1983) [24]," developed by Y. X. Peng and L. R. Long in 2001 [44], was used. The questionnaire has 39 items divided into five dimensions: self-evaluation, information collection, goal selection, planning, and problem-solving. Participants rated their responses on a 5-point Likert-type scale from 1 (no confidence at all) to 5 (complete confidence). Higher scores indicate higher self-efficacy in career decision-making. The original report showed that the internal consistency of several CDMSE subscales ranged from 0.68 to 0.81, with an overall internal consistency of 0.94. In this study, the internal consistency coefficients for the five subscales were 0.88, 0.93, 0.92, 0.91, and 0.87, respectively. The overall internal consistency of the total scale was notably high, with a coefficient of 0.98.

**2.2.4. The Job search clarity Scale.** The Job Search Clarity Scale (JSC), initially developed by Côté et al. (2006) [31] and revised by Wenhua Liao in 2007 [45], was used. The scale consists of 5 items and uses a 5-point Likert-type scale ranging from 1 (does not describe me at all) to 5 (describes me well). Higher scores indicate higher levels of job search clarity. In this study, the JSC had good internal consistency with a reliability coefficient of 0.74.

**2.2.5. The Proactive Personality Scale.** The Proactive Personality Scale (PPS), originally developed by Bateman and Crant (1993) [36] and revised for the Chinese context by Chia-Yin Shang and Yi-Qun Gan in 2009 [46], includes 11 items. It uses a 7-point Likert-type response format from 1 (strongly disagree) to 7 (strongly agree). The scale

assesses proactive personality traits, reflecting individuals' tendencies to take initiative, act independently, and influence environmental changes. In this study, the Chinese version of the PPS had good internal consistency with a reliability coefficient of 0.87.

## 2.3. Statistical analysis

All statistical analyses were conducted using IBM SPSS 21.0 and Mplus 8.3. Prior to hypothesis testing, data were screened for missing values, outliers, and multivariate normality. Missing data patterns were examined using Little's MCAR test. Given that missing data accounted for less than 5% of responses and were determined to be missing completely at random, listwise deletion was employed for primary analyses. Univariate outliers (values exceeding ±3.29 standard deviations) and multivariate outliers were identified and retained after confirming they represented legitimate extreme responses rather than data entry errors. The statistical analysis for this study was conducted through several steps, including confirmatory factor analysis (CFA) to assess discriminant validity, structural equation-based mediation analysis to examine mediating effects, and a moderated mediation model test using the latent moderated structural equations (LMS) approach.

The detailed steps are as follows: Confirmatory factor analysis (CFA) was employed to assess the discriminant validity of the study variables. Following the recommendation of [47] for enhanced reliability and model fit through error reduction, we utilized the technique of parceling. Proactive personality was divided into three indicators based on the "item-to-construct balance" determined by factor loadings. Social support, CDSE, and CDMD were parceled into three, five, and three indicators, respectively, using the "internal consistency" criterion derived from subscale mean scores. Job search clarity was represented by measured items, with one item having a factor loading less than 0.4 being excluded, while the remaining four measured items were directly incorporated into the measurement model. Model fit was evaluated using indices such as the Comparative Fit Index (CFI), Tucker-Lewis Index (TLI), Root Mean Square Error of Approximation (RMSEA), and Standardized Root Mean Square Residual (SRMR). A model was considered to have a good fit if CFI and TLI exceeded 0.90, and SRMR and RMSEA were below 0.08, in accordance with the criteria proposed by [48].

In line with the testing strategy of Wen and Ye (2014) [49], we conducted structural equation-based mediation analysis to examine the mediating effects of CDSE and job search clarity in a two-step process. First, the structural equation model from the independent variable to the dependent variable was assessed. Subsequently, in the second step, the structural equation model with the inclusion of mediator variables was tested. Notably, the bias-corrected (BC) bootstrap approach was used to estimate confidence intervals for the mediating effects.

We performed a moderated mediation model test using the latent moderated structural equations (LMS) approach as recommended by Fang and Wen (2018) [50]. Recognizing that multiple linear regression analysis tends to overlook measurement error, leading to underestimation of mediating and moderating effects [51], our approach, based on latent variables, allowed for effective control over measurement error, resulting in a more accurate estimation of mediating and moderating effects [50]. The LMS approach involved a three-step process to test the moderated mediation model. Initially, the baseline structural equation modeling (SEM) model (M0) without the latent interaction term was estimated. Subsequently, the structural model (M1) with the latent interaction term was estimated in the second step. In the third step, moderated mediation effect analysis was conducted using the coefficient multiplicative method. If the confidence interval does not include 0, it indicates that the moderated mediation effect is significant. If the confidence interval does not include 0, it indicates that the moderated mediation effect is significant. Since the LMS approach does not provide traditional model fit indices, we utilized the log-likelihood ratio test and the Akaike Information Criterion (AIC) value for estimation. The test statistic $D = -2 * (LM0-LM1)$ was calculated for the log-likelihood ratio test based on the H0 value in the results of the Mplus run. A significant D-value chi-square test, when the baseline model fits well, indicates that the moderated mediation model fits better [52,53]. To reduce the impact of multicollinearity, all indicators of latent variables were standardized, and gender and grade were included as control variables.

## 3. Results

### 3.1. Common method bias and discriminant validity tests

Given the reliance on self-report data collection, this study acknowledges the potential for common method biases. To mitigate this concern, stringent procedural controls were implemented during questionnaire administration, emphasizing anonymity, result confidentiality, and the exclusive use of data for academic research. Further, an examination of common method bias was conducted during data analysis. Initially, Harman's one-factor test was applied, revealing that 12 eigenvalues exceeded 1. However, the variance explained by the first factor was 31.34%, falling below the critical criterion of 40% [54]. Additionally, a one-way model test employing confirmatory factor analysis assessed model fit. Results indicated suboptimal fit with $\chi 2 = 4949.54$, $df = 135$, RMSEA = 0.19, SRMR = 0.14, CFI = 0.69, and TLI = 0.65. This outcome suggests that common method bias is not a significant concern in this study. Discriminant validity was assessed using confirmatory factor analysis (CFA) for social support, CDSE, job search clarity, CDMD, and proactive personality. A series of embedded models were compared, and the results in Table 1 demonstrate that the five-factor model exhibited the best fit ($\chi 2 = 383.43$, $df = 125$, RMSEA = 0.05, SRMR = 0.04, CFI = 0.98, TLI = 0.98), affirming strong discriminant validity among the primary variables.

### 3.2. Descriptive statistical results and correlation analysis

The sample consisted of 651 male students (65.69%) and 340 female students (34.31%), with 569 freshmen (57.42%), 328 sophomores (33.09%), and 94 juniors (9.49%). The correlation analysis results, presented in Table 2, reveal significant associations among various study variables. Social support, CDSE, job search clarity, and proactive personality exhibited significant negative correlations with CDMD. Moreover, CDSE, job search clarity, and proactive personality were found to be significantly and positively correlated with social support. Additionally, CDSE, job search clarity, and proactive

**Table 1. Fit Indices of Nested Models.**

| Model | Factors | χ2 | df | RMSEA | SRMR | CFI | TLI |
|---|---|---|---|---|---|---|---|
| Five-factor | A, B, C, D, E | 383.43 | 125 | 0.05 | 0.04 | 0.98 | 0.98 |
| Four-factor | A+B, C, D, E | 1876.29 | 129 | 0.12 | 0.1 | 0.89 | 0.87 |
| Three-factor | A+B+C, D, E | 2754.26 | 132 | 0.14 | 0.11 | 0.83 | 0.8 |
| Two-factor | A+B+C+D, E | 3815.87 | 134 | 0.17 | 0.13 | 0.76 | 0.73 |
| One-factor | A+B+C+D+E | 4949.54 | 135 | 0.19 | 0.14 | 0.69 | 0.65 |

*Note*: Social support = A, career decision-making self-efficacy = B, job search clarity = C, proactive personality = D, career decision-making difficulties = E.

**Table 2. Means, Standard Deviations, and Correlation Coefficients of Variables.**

| Variable | *M* | SD | 1 | 2 | 3 | 4 | 5 | 6 | 7 |
|---|---|---|---|---|---|---|---|---|---|
| 1 Gender | __ | __ | 1.00 | | | | | | |
| 2 Grade | __ | __ | −.055 | 1.00 | | | | | |
| 3 Social Support | 4.77 | 0.95 | .024 | −.032 | 1.00 | | | | |
| 4 Career Decision-Making Self-Efficacy | 3.35 | 0.69 | −.085** | −.006 | .429** | 1.00 | | | |
| 5 Job Search Clarity | 3.31 | 0.61 | .004 | .074* | .382** | .545** | 1.00 | | |
| 6 Proactive Personality | 5.31 | 0.84 | .042 | −.009 | .413** | .474** | .338** | 1.00 | |
| 7 Career Decision-Making Difficulties | 4.73 | 1.10 | −.065* | −.033 | −.322** | −.349** | −.532** | −.265** | 1.00 |

*Note*: $N = 991$, *$p < 0.05$, **$p < 0.01$.

personality displayed positive and significant correlations. Furthermore, CDSE demonstrated a positive correlation with job search clarity. Given the significant correlation between gender and CDMD, a gender difference analysis was conducted. The findings indicated that, concerning the overall career decision-making difficulty score, male students ($N = 651$, $M = 4.78$, $SD = 1.08$) exhibited a significantly higher level of CDMD compared to female students ($N = 340$, $M = 4.63$, $SD = 1.13$), with $t = 2.06$, $p < 0.05$. On the lack of information subscale, male students ($M = 4.68$, $SD = 1.42$) displayed a significantly higher level of CDMD than female students (M = 4.48, $SD = 1.55$), with $t = 2.04$, $p < 0.05$. Although no significant difference was observed on the subscales of lack of preparation and inconsistent information, male students still scored slightly higher than their female counterparts.

### 3.3. Test of the mediating role of career decision-making self-efficacy and job search clarity

Following the test procedure recommended by Zhonglin Wen and Baojuan Ye [49], a two-step analysis was conducted to examine the mediation effect of CDSE and job search clarity. Before conducting the mediation analysis, we assessed the assumptions underlying the mediation model. The assumptions of normality and homogeneity of variance were examined. The Shapiro-Wilk test indicated that the residuals of all variables were approximately normally distributed ($p > 0.05$). Levene's test was used to check the homogeneity of variance, and no significant differences were found in the variances of the outcome variable across different levels of the predictor variables ($p > 0.05$), suggesting that the assumptions for mediation analysis were met.

First, the direct effect of social support on CDMD was tested. The results showed a well-fitting model ($\chi 2$ (18, $N = 991$) = 56.24, $RMSEA = 0.05$, $SRMR = 0.04$, $CFI = 0.99$, $TLI = 0.98$). After controlling for gender and grade, social support significantly and negatively predicted CDMD ($\beta = -0.40$. $SE = 0.03$, $p < 0.001$, 95% $CI$ [−0.46, −0.34]), explaining 16.4% of the variance. Thus, Hypothesis 1 was supported.

In the second step, CDSE and job search clarity were added as mediators to the original model. The new model also showed a good fit ($\chi 2$ (112, $N = 991$) = 375.93, $RMSEA = 0.05$, $SRMR = 0.05$, $CFI = 0.98$, $TLI = 0.98$). The variance explained in CDSE, job search clarity, and CDMD were 20.6%, 48.3%, and 34.2%, respectively. The results of the mediation effect analysis are presented in Fig 2 and Table 3, including point estimates, standard errors, 95% bias-corrected bootstrap confidence intervals, and relative mediation effects (percentage of total effect).

The results showed that social support significantly and positively predicted job search clarity ($\beta = 0.18$, $SE = 0.04$, $p < 0.001$, 95% $CI$ [0.12, 0.25]), and job search clarity significantly and negatively predicted CDMD ($\beta = -0.45$, $SE = 0.05$, $p < 0.001$, 95% $CI$ [−0.54, −0.37]). Social support had a significant total effect on CDMD ($\beta = -0.17$, $SE = 0.04$, $p < 0.001$, 95% $CI$ [−0.25, −0.10]). Job search clarity partially mediated the relationship between social support and CDMD, with an indirect effect of −0.08, 95% CI [−0.12, −0.05], accounting for 20% of the total effect, supporting Hypothesis 2b. The relative mediation effect of 20% was calculated by dividing the indirect effect (−0.08) by the total effect (−0.40) and multiplying by 100%.

In addition, social support significantly and positively predicted CDSE ($\beta = 0.45$, $SE = 0.03$, $p < 0.001$, 95% $CI$ [0.39, 0.51]), and CDSE significantly and positively predicted job search clarity ($\beta = 0.59$, $SE = 0.03$, $p < 0.001$, 95% $CI$ [0.53, 0.65]). A serial mediation effect from social support to CDMD via CDSE and job search clarity was observed, with a serial mediation indirect effect of −0.12, 95% CI [−0.16, −0.09], accounting for 30% of the total effect, supporting Hypothesis 2c. The relative mediation effect of 30% was calculated by dividing the serial indirect effect (−0.12) by the total effect (−0.40) and multiplying by 100%.

However, CDSE was found to have no direct effect on CDMD ($\beta = -0.04$, $SE = 0.04$, $p > 0.05$, 95% $CI$ [−0.12, 0.05]), not supporting Hypothesis 2a. Nevertheless, 95% confidence intervals for all indirect paths, except the path of social support to CDMD via CDSE, did not include 0, and indirect effects reached the level of significance. To provide further confidence in the proposed relationships, a goodness-of-fit test for the structural model was conducted. The results indicated that the structural model exhibited a good fit to the data ($\chi^2$ (112, $N = 991$) = 375.93, RMSEA = 0.05, SRMR = 0.05, CFI = 0.98, TLI = 0.98), suggesting that the proposed model adequately represents the relationships among the variables.

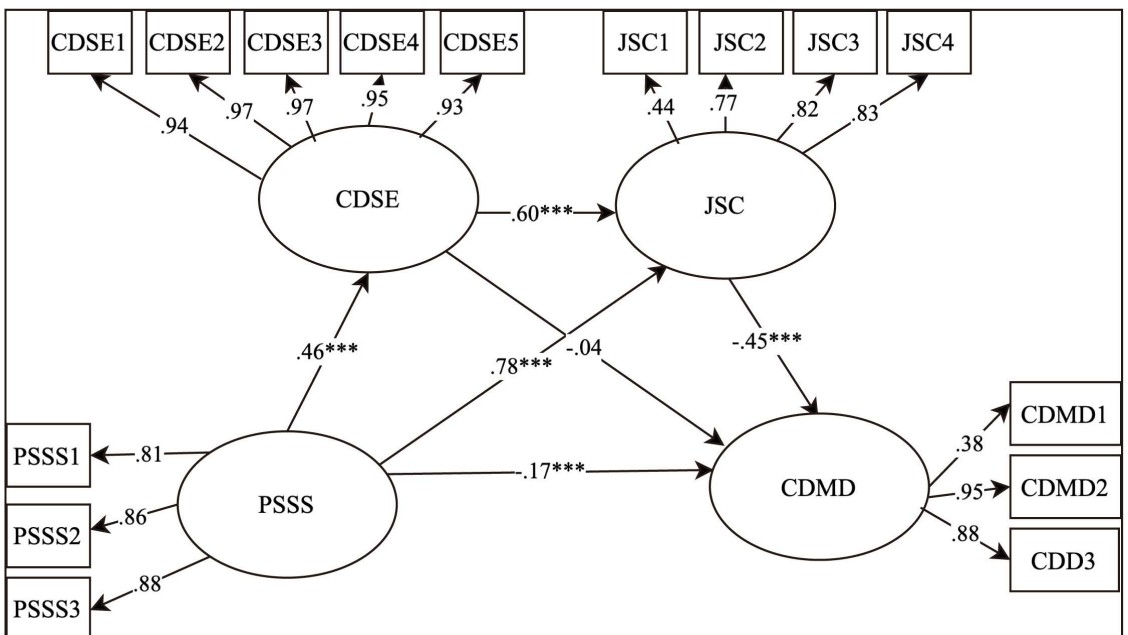

**Fig 2. The results of the mediation effect analysis.** PSSS = Perceived Social Support; JSC = Job Search Clarity; CDSE = Career Decision Self-Efficacy; CDMD = Career Decision-making Difficulties.

**Table 3. Mediation Effect Analysis of Career Decision Self-Efficacy and Job Search Clarity on CDMD.**

| Mediation Model | β | S.E. | Lower 2.5% | Upper 2.5% | Relative Mediation Effect |
|---|---|---|---|---|---|
| Total Indirect Effect | −0.22 | 0.03 | −0.28 | −0.18 | 55% |
| PSSS-JSC-CDMD | −0.08 | 0.02 | −0.12 | −0.05 | 20% |
| PSSS-CDSE-JSC-CDMD | −0.12 | 0.02 | −0.16 | −0.09 | 30% |
| PSSS-CDSE-CDMD | −0.02 | 0.02 | −0.06 | 0.02 | 5% |

*Note*: PSSS = Perceived Social Support; JSC = Job Search Clarity; CDSE = Career Decision Self-Efficacy; CDMD = Career Decision-making Difficulties; Lower 2.5% = Lower limit of 95% Bootstrap Confidence Interval; Upper 2.5% = Upper limit of 95% Bootstrap Confidence Interval. Relative mediation effects were calculated as (indirect effect/total effect) × 100%.

### 3.4. Test of moderated mediating role of proactive personality

To examine the moderated mediation model, we utilized a three-step Latent Moderated Structural (LMS) approach. Before estimating the models, the assumptions for the moderation analysis were assessed. The interaction terms were examined for multicollinearity, and the variance inflation factors (VIFs) were all below 10, indicating that multicollinearity was not a significant issue. For this analysis, two models were estimated. Model 0 included the main effect of proactive personality in the mediation model. The mediation effect analysis indicated that CDSE did not mediate the relationship between social support and CDMD. Consequently, only the indirect path from social support to CDMD via job search clarity was considered for moderation by proactive personality.

Model 0 exhibited good fit indices: $\chi^2$ (162, $N$ = 991) = 570.20, $RMSEA$ = 0.05, $SRMR$ = 0.08, $CFI$ = 0.98, and $TLI$ = 0.97. In Model 1, a latent interaction term was introduced. The log-likelihood ratio test and AIC values were employed to compare Model 1 to Model 0. The results showed that Log-Likelihood$_{M0}$ = −17771.11, Log-Likelihood$_{M1}$ = −17754.88,

$D = 32.46$, $\triangle df = 1$, $p < 0.001$. Additionally, the AIC value of Model 1 (35637.76) was lower than that of Model 0(35668.22), indicating a significant improvement in model fit by incorporating the interactions ($\beta = -0.17$, $SE = 0.04$, $p < 0.001$). Thus, proactive personality moderates the latter part of the mediation effect (see Fig 3).

Proactive personality significantly moderated the indirect path from social support to CDMD via job search clarity ($\beta = -0.04$, $SE = 0.01$, $p < 0.001$, 95% $CI$ [−0.06, −0.02]) and the serial mediation indirect path from social support to CDMD via CDSE and job search clarity ($\beta = -0.06$, $SE = 0.01$, $p < 0.001$, 95% $CI$ [−0.08, −0.04]). Therefore, Hypothesis 3b was supported, while Hypothesis 3a was not.

To further elucidate the interaction, simple slope analysis was conducted, and interactions were plotted in Fig 4. The results revealed that under low levels of proactive personality ($Z = -1$), CDMD displayed a significant downward trend as job search clarity increased ($\beta = -0.21$, $SE = 0.05$, $p < 0.001$, 95% $CI$ [−0.32, −0.11]). Similarly, above high levels of proactive personality ($Z = 1$), CDMD exhibited a significant downward trend as job search clarity increased ($\beta = -0.54$, $SE = 0.08$, $p < 0.001$, 95% $CI$ [−0.72, −0.41]). This suggests that the effect of job search clarity on reducing career decision-making difficulties became more pronounced as proactive personality increased.

## 4. Discussion

The relationship between social support and CDMD has garnered attention in previous studies. However, the direct link between social support and CDMD among vocational college students, along with the underlying mechanisms, remains underexplored. Grounded in the social cognitive model of CSM, this study introduced two mediating variables: CDSE and job search clarity and one moderating variable: proactive personality. By constructing and testing a moderated mediating model, we obtained several meaningful findings. First, the overall CDMD level among vocational students was moderate, with male students exhibiting significantly higher CDMD than female students. Second, social support negatively predicted

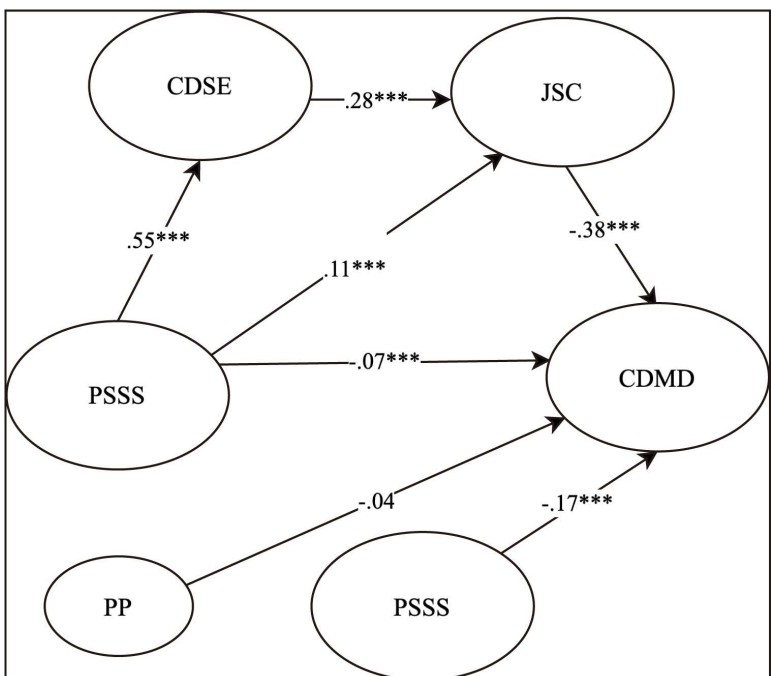

**Fig 3. Results of the Moderation Effect Test.** PSSS = Perceived Social Support; JSC = Job Search Clarity; CDSE = Career Decision Self-Efficacy; CDMD = Career Decision-making Difficulties; PP = Proactive Personality.

  

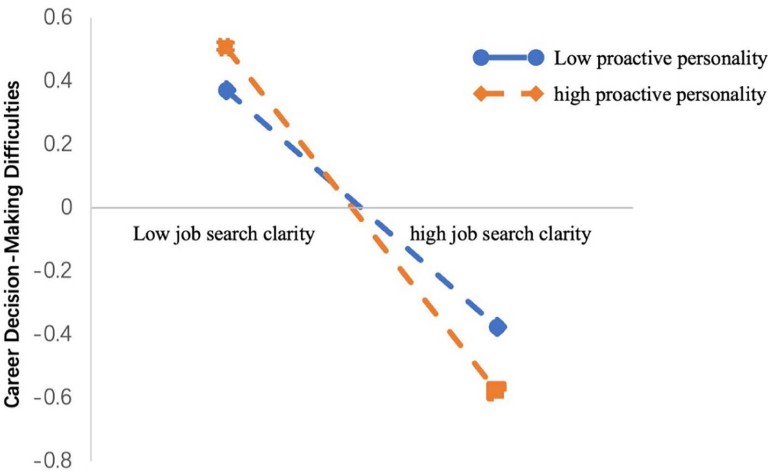

**Fig 4. The Moderating Role of Proactive Personality in the Relationship between Job Search Clarity and CDMD.** Simple slope analysis shows: low proactive personality (β = −0.21, SE = 0.05, 95% CI [−0.32, −0.11]); high proactive personality (β = −0.54, SE = 0.08, 95% CI [−0.72, −0.41]).

CDMD, indicating that students with higher perceived social support encountered fewer difficulties in career decisions. Third, mediation analyses revealed that job search clarity partially mediated the relationship between social support and CDMD, accounting for 20% of the total indirect effect. Finally, proactive personality moderated the indirect effects, enhancing the negative relationship between job search clarity and CDMD.

### 4.1. Characteristics of career decision-making difficulties of vocational college students

The study revealed that the CDMD of vocational college students were generally at a medium level ($M = 4.73$, $SD = 1.10$). Further examination of the group differences in CDMD among vocational college students showed a significant association with gender. Male students reported significantly higher levels of CDMD than female students. This aligns with prior research indicating that gender can influence career decision-making processes and outcomes, highlighting the importance of considering gender-specific factors in vocational guidance interventions [55]. In the Chinese cultural context, men face greater pressure and social expectations in career development, leading them to consider more factors when making career choices, especially salary and promotion opportunities. On the other hand, due to the long-term male-dominated job market, women tend to exhibit more exploratory and preparatory behaviors in the job-seeking process. They are more likely to prioritize stability over salary when making career choices. Therefore, male students in higher vocational education face a higher level of difficulties in the career decision-making process compared to female students. This aligns with prior research indicating that gender can influence career decision-making processes and outcomes, highlighting the importance of considering gender-specific factors in vocational guidance interventions.

### 4.2. The relationship between social support and career decision-making difficulties

The CSM emphasizes the crucial role of environmental support and hindrances in individuals' career decision-making processes. Adequate environmental support can drive individuals to make adaptive career decisions. Previous studies have shown a positive association between social support and CDMD [26,56]. Consistent with these findings, this study also found that social support significantly and negatively predicted CDMD (β = −0.40, p < 0.001, 95% CI [−0.46, −0.34]), explaining 16.4% of the variance among vocational college students. Students with higher levels of social support experienced lower CDMD. This result reaffirms the positive role of social support in reducing CDMD and highlights its importance for vocational college students. Our findings echo the work of prior researchers who have established the supportive

role of social networks in career development, thereby reinforcing the theoretical foundations of the CSM model within the specific context of vocational education.

### 4.3. The mediating role of career decision-making self-efficacy, job search clarity

This study found that social support directly predicted CDSE. These results suggest that social support operates through multiple pathways to influence career decision-making, and that enhancing social support systems may be a valuable strategy for improving vocational outcomes among students. As a significant environmental factor, vocational college students who received more social support had higher levels of CDSE, while those with low social support had lower levels. This is in line with previous research [22,26]. In the Chinese cultural context, college students' career choices are often influenced by significant others such as parents, friends, and teachers. Given that vocational college students do not have the advantage of educational qualifications, they rely not only on their own professional abilities but also on the support of others. The more resources and support they received, the more confident they were in making suitable career choices. Additionally, social support significantly and positively predicted job search clarity. Higher levels of social support led to greater goal clarity for future careers, while lower levels resulted in lower job search clarity, which is consistent with the hypothesis of the social cognitive model of CSM. These results suggest that social support operates through multiple pathways to influence career decision-making, and that enhancing social support systems may be a valuable strategy for improving vocational outcomes among students.

According to the CSM, social support can influence career decision-making through CDSE and job search clarity, as well as through a continuous path of CDSE and job search clarity to affect career decision-making behavior. CDSE is a significant predictor of CDMD [5,29]. Based on the theoretical hypotheses of the social cognitive model of CSM, we explored the underlying mechanisms by which social support influences CDMD, examining the mediating role of CDSE and job search clarity. The study found that job search clarity was modestly and negatively related to CDMD and had a significant direct predictive effect on CDMD. Vocational college students with higher job search clarity had clearer employment goals and were more focused when faced with career choices. They could devote limited time and energy to the most effective job-search behaviors [37], used more job-search resources [32], and experienced fewer CDMD. In contrast, individuals with unclear job search goals would struggle to identify the right job and make adaptive career choices, leading to more time spent exploring different options and increased CDMD [37]. This finding empirically confirms the theoretical hypothesis of the social cognitive model of CSM that job search clarity mediates the relationship between social support and CDMD. In this study, the indirect effect from social support to CDMD via job search clarity accounted for 20% of the total indirect effect. These results underscore the practical importance of fostering job search clarity as a means of reducing CDMD among vocational students.

Interestingly, our study found that the direct predictive effect of CDSE on CDMD was not significant, meaning that CDSE did not mediate the relationship between social support and CDMD. Although this finding is inconsistent with the theoretical hypothesis, it is supported by some previous studies [19,57]. A foreign longitudinal study of adolescents also found that while CDSE was significantly related to CDMD at the same time point, changes in CDSE over time did not predict changes in CDMD [58]. These findings suggest that CDSE is not a direct antecedent variable of CDMD and that there may be a third variable mediating the relationship between them [58]. In our study, CDSE was significantly correlated with CDMD. The social cognitive model of CSM suggests that goals may play such a role. Goals can either directly influence career decision-making behavior or act as a bridge between social support, self-efficacy, and career decision-making behavior. The study found that CDSE significantly and positively predicted job search clarity. Vocational college students with higher CDSE had more confidence in their ability to complete tasks related to career decision-making and thus had a clearer understanding of their career interests, strengths, and goals [59]. This confirms the hypothesis of the social cognitive model of CSM that job search clarity acts as a bridge between CDSE and CDMD. This indicates that future research and interventions might benefit from exploring additional variables that could elucidate the relationship between self-efficacy and CDMD.

Furthermore, this study found a serial mediation effect of CDSE and job search clarity between social support and CDMD. Vocational college students with higher levels of social support had more confidence in making suitable career choices, clearer future career goals, and thus faced fewer difficulties in making career decisions. The serial mediation effect of CDSE and job search clarity accounted for 30% of the total indirect effect, exceeding the value of the mediation effect of job search clarity alone. This finding reveals that although CDSE cannot directly affect the level of CDMD, it can indirectly influence it by increasing job search clarity. Therefore, higher education institutions and career counselors should enhance social support for vocational college students through various channels and improve students' CDSE and job search clarity through individual career counseling, group counseling, and career guidance courses to reduce the level of CDMD. Such strategies could include organizing workshops aimed at building self-efficacy and providing resources that enhance job search clarity, thereby equipping students with the tools needed to navigate their career decisions more effectively.

### 4.4. The moderating role of proactive personality

The social cognitive model of CSM posits that personality factors can facilitate or hinder goal and career behavior outcomes. This study examined the effect of proactive personality on the relationship between job search clarity and CDMD. The results showed that proactive personality was a protective factor for CDMD among vocational college students, consistent with previous findings [60–62]. More importantly, proactive personality moderated the indirect path from social support to CDMD via job search clarity and the serial mediation indirect path from social support to CDMD via CDSE and job search clarity. It had a moderating effect on the relationship between job search clarity and CDMD. Proactive personality had a "reinforcing" effect on the weakening effect of job search clarity on CDMD, making the negative prediction of job search clarity on CDMD more significant for vocational college students with higher levels of proactive personality than for those with lower levels. This finding is consistent with the interpretation that students who are more proactive may be better able to leverage the benefits of job search clarity, leading to fewer CDMD. Practitioners might consider incorporating personality assessments to identify students who could benefit from targeted interventions aimed at enhancing proactivity.

The social cognitive model of CSM can explain the moderating role in which the combination of individual differences and social environmental factors determines career behavior rather than acting as separate influences [13]. College students with higher levels of proactive personality are more likely to engage in career exploration behaviors, have a better understanding of themselves and the external career environment, and have a clearer understanding of the direction and goals of their career development [63]. When faced with career choices, they can choose the right career according to their career goals and interests and experience fewer decision-making difficulties. Through this process, high proactive personality reinforces the negative predictive effect of job search clarity on CDMD.

#### 4.4.1. Limitations and future research directions.
Notwithstanding its substantive contributions to understanding CDMD among vocational college students, this investigation is circumscribed by several methodological and conceptual limitations that warrant deliberate attention in subsequent scholarship. Particular exigencies surrounding the cross-sectional design merit prioritized resolution to advance the epistemological foundations of this domain.

Primarily, the exclusive dependence on monomethod self-report measures introduces non-trivial risks of response bias and common method variance—a psychometric artifact potentially confounding observed relationships. Future research should circumvent this constraint through triangulated assessment protocols, incorporating multi-informant data (e.g., parental, pedagogical, or employer perspectives) to cultivate richer ecological validity and attenuate measurement artifact contamination. Complementarily, implementing temporal separation of predictor-criterion measurements across distinct assessment waves would further dismantle method covariance while fortifying inferential robustness.

Secondly, the inherent constraints of cross-sectional topology fundamentally preclude causal attribution among examined constructs. While statistically significant associations emerged linking social support, CDSE, job search clarity, proactive personality, and CDMD, these relational patterns demand circumspect interpretation absent temporal precedence

evidence. Future studies should adopt longitudinal or experience-sampling methodologies to elucidate causal sequences, developmental cascades, and reciprocal dynamics among these variables.

Thirdly, the study sample was drawn exclusively from Chinese vocational college students, limiting sample diversity. While this deliberate sampling frame enables contextual depth, it constrains population generalizability. To transcend these geographical and institutional boundaries, future inquiries should incorporate comparative samples spanning diverse educational echelons (e.g., undergraduate, postgraduate) and cross-national vocational training ecosystems. Such strategic expansion would critically interrogate the external validity frontier while delineating universal versus context-bound manifestations of CDMD.

Fourthly, the counterintuitive null finding regarding CDSE 's direct predictive utility contravenes established theoretical postulates, signaling latent mechanistic complexity. This empirical anomaly necessitates rigorous deconstruction of mediating pathways—potentially involving vocational identity crystallization or goal-regulation architectures—that may obfuscate the CDSE – CDMD nexus. Exploring additional psychological variables and contextual moderators could clarify these complex relationships.

Finally, the conspicuous omission of cultural-psychological determinants represents a critical lacuna. Within China's collectivist milieu, familial expectations, sociostructural prescriptions, and gendered occupational norms profoundly scaffold developmental trajectories of career volition. Future research must therefore: (a) systematically integrate cultural moderators (e.g., filial piety, gender role ideology) to map their contingent interactions with psychological antecedents of CDMD; and (b) launch coordinated cross-cultural programmes contrasting individualist versus collectivist vocational ecosystems. Such initiatives would disentangle cultural universals from idiosyncrasies, forging culturally-attuned theoretical frameworks.

## 5. Conclusions

In conclusion, this study, anchored in the social cognitive model of CDSE, thoroughly explored the complex interplay between social support and vocational college students' CDMD. The key findings highlight several vital aspects.

Firstly, social support was identified as a robust direct predictor, negatively impacting vocational college students' CDMD. This finding emphasizes the crucial role of a supportive environment in alleviating the challenges associated with career choices among this student group.

Secondly, the study unveiled the intricate role of job search clarity as a partial mediator in the relationship between social support and CDMD. This adds a layer of detail to our understanding of the pathways through which social support influences vocational college students' career decisions.

Furthermore, the identification of CDSE and job search clarity as serial mediators provides further insight into the complex mechanisms at work. These findings clarify the sequential interplay of psychological processes that link social support to reduced CDMD.

Lastly, the study revealed the moderating effect of proactive personality on the relationship between job search clarity and CDMD. This highlights the significance of individual differences in shaping how vocational college students navigate challenges in their career decision-making journey.

Collectively, these results validate key propositions of the CSM model and carry practical implications. Interventions aiming to enhance social support, promote job search clarity, and strengthen proactive personality traits are recommended to alleviate CDMD among vocational college students. By integrating these strategies into career counseling and educational programs, practitioners can better support students in achieving adaptive and confident career decision-making.

## Author contributions

**Conceptualization:** Rong Chen.

**Data curation:** Rong Chen.

**Formal analysis:** Rong Chen.

**Investigation:** Rong Chen, Qin Zhang.

**Methodology:** Rong Chen.

**Supervision:** Yunfei Cao.

**Writing – original draft:** Rong Chen.

**Writing – review & editing:** Yunfei Cao.

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
