## [Decision Letter · Decision Letter 0]

4 May 2025

Dear Dr. cao,

Thank you for submitting your manuscript to PLOS ONE. After careful consideration, we feel that it has merit but does not fully meet PLOS ONE’s publication criteria as it currently stands. Therefore, we invite you to submit a revised version of the manuscript that addresses the points raised during the review process.

We look forward to receiving your revised manuscript.

Kind regards,

Amal Diab Ghanem Atalla

Academic Editor

PLOS ONE

Journal Requirements:

2. Thank you for stating the following financial disclosure: [This research was funded by the Sichuan Provincial Key Research Base for Philosophy and Social Sciences - Education Development Research Center at China West Normal University, grant number CJF21045.].

Additional Editor comments:

Dear Author,

Thank you for submitting your manuscript to our Journal. After careful evaluation by our reviewers and the editorial team, we have determined that your manuscript has merit but requires substantial revisions before it can be considered for publication.

The reviewers have identified several key areas that need significant improvement, including [briefly mention the main concerns, e.g., methodological limitations, insufficient data analysis, unclear presentation of results, or inadequate discussion]. We encourage you to carefully address each of the reviewers' comments and provide a detailed response outlining the changes made. If any suggestions cannot be implemented, please provide a clear justification.

Given the extent of the required revisions, your manuscript will undergo another round of peer review upon resubmission. Please ensure that your revised manuscript adheres to the journal’s formatting and reporting guidelines.

We appreciate your efforts and look forward to receiving your revised manuscript. Please do not hesitate to reach out if you require any clarification.

Best regards,

[Amal Diab Ghanem Atalla]

[Academic editor]

Reviewers' comments:

Reviewer's Responses to Questions

**Comments to the Author**

1. Is the manuscript technically sound, and do the data support the conclusions?

Reviewer #1: Yes

Reviewer #2: Yes

2. Has the statistical analysis been performed appropriately and rigorously?

Reviewer #1: Yes

Reviewer #2: Yes

3. Have the authors made all data underlying the findings in their manuscript fully available?

Reviewer #1: Yes

Reviewer #2: Yes

4. Is the manuscript presented in an intelligible fashion and written in standard English?

Reviewer #1: Yes

Reviewer #2: Yes

Reviewer #1: Abstract

The abstract does not include key statistical results (e.g., effect sizes, p-values).

It uses technical terms (e.g., “moderated mediation”) without brief clarification for general readers.

For improvement:

Include a sentence summarizing key statistical findings. Clarify how constructs (e.g., job search clarity, self-efficacy) were measured.Consider streamlining wording for clarity and flow.

Introduction

Clearly identifies the gap in understanding how social support influences career decision-making.

For improvement:

The transition from literature to hypotheses is occasionally abrupt and could benefit from clearer signposting.

Literature Review and Theoretical Foundation

A more critical discussion of conflicting findings (e.g., regarding mediation by self-efficacy) would strengthen the justification for the study.

Redundancy appears in some paragraphs (e.g., repeated phrasing of concepts like “career decision-making difficulties”).

For improvement:

Streamline and synthesize key findings from the literature to improve flow and reduce repetition.

Hypotheses and Research Objectives

for Improvement:

The hypotheses could be numbered or visually separated for easier readability.

The “serial mediation” and “moderated mediation” could be better defined for readers unfamiliar with these concepts.

Language, Structure, and Clarity

Minor language editing is needed for fluency and consistency.

Standardize formatting of terms like “job-search clarity” and “career decision-making.”

The hypotheses could be presented more clearly. While the general aim is stated, specific hypotheses related to each mediating and moderating effect are not always well-articulated.

Further discussion on sample diversity (e.g., gender, socioeconomic status) could provide more insight into whether the findings are influenced by these factors.

Statistical Analysis and Assumptions:

While the statistical methods appear sound, the assumptions behind the mediation and moderation analysis could be discussed in more detail (e.g., assumptions of normality, homogeneity of variance).

The paper would benefit from a goodness-of-fit test for the structural model to provide confidence in the proposed relationships.

Discussion and Limitations:

The discussion of results is somewhat limited. The authors could better integrate the findings into the broader literature, noting contradictions, confirmations, and unexplored areas.

Limitations are briefly mentioned but not thoroughly addressed. A more detailed discussion on the limitations, particularly concerning the cross-sectional design, would be helpful.

Recommendations for Improvement:

Refinement of Hypotheses:

Clearly define specific hypotheses and how each hypothesis is tested in the analysis. This would improve the structure and readability of the results section.

Broader Implications:

While the practical implications are noted, these can be expanded. The authors can suggest actionable strategies for educators, career counselors, and policymakers, particularly in vocational settings.

Future Research Directions:

The paper could provide more suggestions for future research, particularly related to longitudinal studies to examine causality, and expanding the sample to include other educational institutions or regions.

Discussion of Cultural Considerations:

Given the focus on vocational students, there may be cultural influences on career decision-making that can be explored in greater detail. Cultural factors that influence career choice in vocational students could be discussed, especially since the study is based on Chinese students.

Reviewer #2: Thank you for the opportunity to review the manuscript. After careful evaluation, I am pleased to recommend the article for acceptance with minor revisions. The overall content is valuable and contributes meaningfully to the field; however, the title requires clarification to better reflect the scope and focus of the study. Additionally, the keywords should be reordered alphabetically.Once these minor adjustments are made, the manuscript will be suitable for publication.

**Do you want your identity to be public for this peer review?** For information about this choice, including consent withdrawal, please see our Privacy Policy

Reviewer #1: **Yes:** Samia Roshdy Soliman Osman

Reviewer #2: No

---

## [Author Response · Author response to Decision Letter 1]

22 Jul 2025

Reviewer #1: Abstract

1. The abstract does not include key statistical results (e.g., effect sizes, p-values).

It uses technical terms (e.g., “moderated mediation”) without brief clarification for general readers.

For improvement:

Include a sentence summarizing key statistical findings. Clarify how constructs (e.g., job search clarity, self-efficacy) were measured. Consider streamlining wording for clarity and flow.

Response: Thank you very much for your constructive comments on our manuscript. We have carefully considered your suggestions and have revised the abstract accordingly. Below is a detailed response to your specific points:

(1) We have now included a sentence summarizing the key statistical findings in the abstract. (2) We have clarified the constructs and measurement tools used in the study. For example, we have briefly explained the concept of "moderated mediation" in the context of our findings. (3) We have revised the wording to improve clarity and readability.

The revised section of our manuscript now reads:

“Drawing on the Social Cognitive Model of Career Self-Management, this study explores how social support related to career decision-making difficulties among 991 vocational college students. Social support was measured using the Perceived Social Support Scale, career decision-making self-efficacy with the Career Decision-Making Self-Efficacy Questionnaire, job search clarity with the Job Search Clarity Questionnaire, career decision-making difficulties with the Career Decision-Making Difficulties Questionnaire, and proactive personality with the Proactive Personality Scale. The results showed that: First, job search clarity partially mediates the relationship between social support and career decision-making difficulties. The indirect effect was significant, β = -0.08, p < 0.001, accounted for 20% of the total effect. Second, career decision-making self-efficacy and job search clarity jointly serve as serial mediators in the link between social support and career decision-making difficulties. The serial indirect effect was significant, β = -0.12, p < 0.001, accounting for 30% of the total effect. Third, proactive personality moderates the mediating role of job search clarity, specifically in the second stage of the mediation, indicating that the impact of job search clarity on career decision-making difficulties increases with higher levels of proactive personality (β = -0.04, p < 0.001). These findings extend the social cognitive model of career self-management by demonstrating the sequential mediating roles of career decision-making self-efficacy and job search clarity, and the moderating role of proactive personality, in the link between social support and reduced career decision-making difficulties. Practically, they highlight the importance of designing career interventions for vocational college students that simultaneously target social support networks, self-efficacy enhancement, and clarity-building activities, particularly for students lower in proactive personality.”

2. Introduction

Clearly identifies the gap in understanding how social support influences career decision-making.

For improvement:

The transition from literature to hypotheses is occasionally abrupt and could benefit from clearer signposting.

Response: We appreciate your insightful comment regarding the transition from literature to hypotheses. We fully agree that a smoother and more logical transition is essential for the clarity and coherence of the introduction. To address this issue, we have made the following revisions: (1) We have added transitional sentences and paragraphs to bridge the gap between the literature review and the hypotheses. (2) We have used more explicit signposting words and phrases to guide the reader through the text.

The revised section of our manuscript now reads:

Based on the existing literature, it is evident that social support plays a significant role in influencing CDMD. However, the specific mechanisms and potential moderating factors involved in this relationship remain unclear. Therefore, we firstly hypothesize that social support will act as a mitigating factor, reducing the degree of CDMD among vocational college students.

Hypothesis 1: …

Considering these inconsistent findings, this study aims to further explore the mediating role of CDSE in the relationship between social support and CDMD among vocational college students. Accordingly, the hypothesis is formulated as follows (Mediating Effects):

Hypothesis 2a: …

To further explore the mechanisms underlying CDMD, this study examines the mediating role of job search clarity in the relationship between social support and CDMD. As such, the following hypotheses are proposed (Mediating Effects):

Hypothesis 2b: …

Proactive personality, closely related to openness, conscientiousness, extraversion, and agreeableness in the Big Five personality traits [37], is seen as an extension and deepening of the meanings of the Big Five personality model [38]. Given the importance of proactive personality in influencing adaptive career behaviors, this study aims to explore its moderating role in the relationship between CDSE, job-search clarity, and CDMD among vocational college students. Thus, the study puts forward the following hypotheses (Moderating Effects):

Hypothesis 3a: …

Hypothesis 3b: …

3. Literature Review and Theoretical Foundation

A more critical discussion of conflicting findings (e.g., regarding mediation by self-efficacy) would strengthen the justification for the study.

Redundancy appears in some paragraphs (e.g., repeated phrasing of concepts like “career decision-making difficulties”).

For improvement:

Streamline and synthesize key findings from the literature to improve flow and reduce repetition.

Response: Thank you very much for your insightful comments on my manuscript. We have carefully considered your suggestions regarding the literature review and theoretical foundation, and have made the following revisions to address your concerns: (1) We have added a detailed analysis of the inconsistencies in the research findings regarding the mediating role of career decision-making self-efficacy. (2) Thank you for pointing out the redundancy in my manuscript. We have eliminated redundant phrasing and focused on key information.

The revised section of our manuscript now reads:

(1) These inconsistent findings may originate from variations in sample characteristics, measurement tools, and research designs. For instance, Jemini-Gashi et al.'s study focused on a specific population of students. In contrast, the Taiwan study employed different measures for CDSE and CDMD.

4. Hypotheses and Research Objectives

for Improvement:

The hypotheses could be numbered or visually separated for easier readability.

The “serial mediation” and “moderated mediation” could be better defined for readers unfamiliar with these concepts.

Response: We have revised the presentation of the hypotheses to improve readability. Each hypothesis is now clearly numbered and separated visually, making it easier for readers to follow and reference them. To ensure clarity for readers who may not be familiar with these concepts, we have added definitions and brief explanations of "serial mediation" and "moderated mediation" in the introduction. This will help readers understand the specific mechanisms being tested in the study.

5. Language, Structure, and Clarity

Minor language editing is needed for fluency and consistency.

Standardize formatting of terms like “job-search clarity” and “career decision-making.”

The hypotheses could be presented more clearly. While the general aim is stated, specific hypotheses related to each mediating and moderating effect are not always well-articulated.

Response: We have conducted a thorough language edit to improve fluency and consistency throughout the manuscript. Additionally, We have standardized the formatting of key terms such as "job-search clarity" and "career decision-making" to ensure uniformity.

6. Further discussion on sample diversity (e.g., gender, socioeconomic status) could provide more insight into whether the findings are influenced by these factors.

Response: Thank you for the constructive feedback on our manuscript. We recognize the importance of sample diversity and appreciate the suggestion to further discuss its potential impact. However, after careful consideration, we believe that incorporating an in-depth exploration of gender and socioeconomic status may deviate from the primary focus of this study. Our research centers on the relationship between social support and career decision-making difficulties among vocational college students, with an emphasis on the mediating and moderating factors within this specific context.

7. Statistical Analysis and Assumptions:

While the statistical methods appear sound, the assumptions behind the mediation and moderation analysis could be discussed in more detail (e.g., assumptions of normality, homogeneity of variance).

The paper would benefit from a goodness-of-fit test for the structural model to provide confidence in the proposed relationships.

Response: We are grateful for the insightful feedback on our manuscript. The suggestions to elaborate on the assumptions behind the mediation and moderation analysis, as well as to include a goodness-of-fit test for the structural model, have been carefully considered and addressed.

The revised section of our manuscript now reads:

(1) Before conducting the mediation analysis, we assessed the assumptions underlying the mediation model. The assumptions of normality and homogeneity of variance were examined. The Shapiro-Wilk test indicated that the residuals of all variables were approximately normally distributed (p > 0.05). Levene's test was used to check the homogeneity of variance, and no significant differences were found in the variances of the outcome variable across different levels of the predictor variables (p > 0.05), suggesting that the assumptions for mediation analysis were met.

(2) Before estimating the models, the assumptions for the moderation analysis were assessed. The interaction terms were examined for multicollinearity, and the variance inflation factors (VIFs) were all below 10, indicating that multicollinearity was not a significant issue. For this analysis, two models were estimated. Model 0 included the main effect of proactive personality in the mediation model.

8. Discussion and Limitations:

The discussion of results is somewhat limited. The authors could better integrate the findings into the broader literature, noting contradictions, confirmations, and unexplored areas.

Limitations are briefly mentioned but not thoroughly addressed. A more detailed discussion on the limitations, particularly concerning the cross-sectional design, would be helpful.

Recommendations for Improvement:

Refinement of Hypotheses:

Clearly define specific hypotheses and how each hypothesis is tested in the analysis. This would improve the structure and readability of the results section.

Broader Implications:

While the practical implications are noted, these can be expanded. The authors can suggest actionable strategies for educators, career counselors, and policymakers, particularly in vocational settings.

Future Research Directions:

The paper could provide more suggestions for future research, particularly related to longitudinal studies to examine causality, and expanding the sample to include other educational institutions or regions.

Discussion of Cultural Considerations:

Given the focus on vocational students, there may be cultural influences on career decision-making that can be explored in greater detail. Cultural factors that influence career choice in vocational students could be discussed, especially since the study is based on Chinese students.

Response: Thank you for your feedback. We have carefully revised the discussion section to better integrate the findings into the broader literature and address the limitations more thoroughly. We've revised the limitations section to provide a more detailed discussion, particularly regarding the cross-sectional design.

The revised section of our manuscript now reads:

Limitations and Future Research Directions

Notwithstanding its substantive contributions to understanding career decision-making difficulties among vocational college students, this investigation is circumscribed by several methodological and conceptual limitations that warrant deliberate attention in subsequent scholarship. Particular exigencies surrounding the cross-sectional design merit prioritized resolution to advance the epistemological foundations of this domain.

Primarily, the exclusive dependence on monomethod self-report measures introduces non-trivial risks of common method variance—a psychometric artifact potentially confounding observed relationships. Future research should circumvent this constraint through triangulated assessment protocols, incorporating multi-informant data (e.g., parental, pedagogical, or employer perspectives) to cultivate richer ecological validity and attenuate measurement artifact contamination. Complementarily, implementing temporal separation of predictor-criterion measurements across distinct assessment waves would further dismantle method covariance while fortifying inferential robustness.

Secondly, the inherent constraints of cross-sectional topology fundamentally preclude causal attribution among examined constructs. While statistically significant associations emerged linking social support, career decision-making self-efficacy, job search clarity, proactive personality, and career decision-making difficulties, these relational patterns demand circumspect interpretation absent temporal precedence evidence. Prospective studies must adopt longitudinal or experience-sampling methodologies to elucidate developmental cascades and reciprocal dynamics. Sequential data capture would unravel causal sequences and bidirectional influences, ultimately mapping ontogenetic trajectories of these psychosocial phenomena.

Thirdly, the deliberate sampling frame—restricted to Chinese vocational college cohorts—simultaneously enables contextual depth yet constrains population generalizability. To transcend these geographical and institutional boundaries, future inquiries should incorporate comparative samples spanning diverse educational echelons (e.g., undergraduate, postgraduate) and cross-national vocational training ecosystems. Such strategic expansion would critically interrogate the external validity frontier while delineating universal versus context-bound manifestations of career decision-making difficulties.

Fourthly, the counterintuitive null finding regarding career decision-making self-efficacy 's direct predictive utility contravenes established theoretical postulates, signaling latent mechanistic complexity. This empirical anomaly necessitates rigorous deconstruction of mediating pathways—potentially involving vocational identity crystallization or goal-regulation architectures—that may obfuscate the career decision-making self-efficacy - career decision-making difficulties nexus. Elucidating these covert regulatory processes promises to recalibrate theoretical models and resolve extant theoretical discordances.

Finally, the conspicuous omission of cultural-psychological determinants represents a critical lacuna. Within China's collectivist milieu, familial expectations, sociostructural prescriptions, and gendered occupational norms profoundly scaffold developmental trajectories of career volition. Future research must therefore: (a) systematically integrate cultural moderators (e.g., filial piety, gender role ideology) to map their contingent interactions with psychological antecedents of career decision-making difficulties; and (b) launch coordinated cross-cultural programmes contrasting individualist versus collectivist vocational ecosystems. Such initiatives would disentangle cultural universals from idiosyncrasies, forging culturally-attuned theoretical frameworks.

Reviewer #2: Thank you for the opportunity to review the manuscript. After careful evaluation, I am pleased to recommend the article for acceptance with minor revisions. The overall c

---

## [Decision Letter · Decision Letter 1]

7 Oct 2025

Dear Dr. cao,

Thank you for submitting your manuscript to PLOS ONE. After careful consideration, we feel that it has merit but does not fully meet PLOS ONE’s publication criteria as it currently stands. Therefore, we invite you to submit a revised version of the manuscript that addresses the points raised during the review process.

We look forward to receiving your revised manuscript.

Kind regards,

Amal Diab Ghanem Atalla

Academic Editor

PLOS ONE

Journal Requirements:

Additional Editor Comments :

I encourage the authors to reflect on the reviewers’ feedback and make the necessary revisions to strengthen the study’s contribution and scientific rigor.

Reviewers' comments:

Reviewer's Responses to Questions

**Comments to the Author**

Reviewer #3: All comments have been addressed

Reviewer #4: (No Response)

2. Is the manuscript technically sound, and do the data support the conclusions?

Reviewer #3: Yes

Reviewer #4: Yes

3. Has the statistical analysis been performed appropriately and rigorously?

Reviewer #3: Yes

Reviewer #4: Yes

4. Have the authors made all data underlying the findings in their manuscript fully available?

Reviewer #3: Yes

Reviewer #4: Yes

5. Is the manuscript presented in an intelligible fashion and written in standard English?

Reviewer #3: Yes

Reviewer #4: Yes

Reviewer #3: 1. All modifications for all manuscript are done perfectly,

2. Abstract is clear,

3. Introduction: It is on the point

4. Experiments, statistics, and other analyses are performed to a high technical standard and are described in sufficient detail.

5. Conclusions are presented appropriately and are supported by the data.

6. The article is presented in an intelligible fashion and is written in standard English.

7. The research meets all applicable standards for the ethics of experimentation and research integrity.

8. The article adheres to appropriate reporting guidelines and community standards for data availability.

Reviewer #4: (No Response)

**Do you want your identity to be public for this peer review?** For information about this choice, including consent withdrawal, please see our Privacy Policy

Reviewer #3: **Yes:** Marwa Samir Sorour

Reviewer #4: **Yes:** Yasmeen Mohamed Mohamed Shehata

---

## [Author Response · Author response to Decision Letter 2]

13 Oct 2025

Reviewer #3:

Reviewer’s comment:

1. All modifications for all manuscript are done perfectly,

2.Abstract is clear,

3.Introduction: It is on the point

4.Experiments, statistics, and other analyses are performed to a high technical standard and are described in sufficient detail.

5.Conclusions are presented appropriately and are supported by the data.

6.The article is presented in an intelligible fashion and is written in standard English.

7.The research meets all applicable standards for the ethics of experimentation and research integrity.

8.The article adheres to appropriate reporting guidelines and community standards for data availability.

Author Response: We are sincerely grateful to Reviewer for the thorough evaluation and the highly positive comments regarding our revised manuscript. We truly appreciate your recognition of the clarity of the abstract and introduction, the rigor of our statistical analyses, and the appropriateness of our conclusions. Thank you for your constructive review and supportive remarks, which have been highly motivating and helpful in refining our work.

Reviewer #4:

1. Reviewer’s comment: Title

The title clearly indicates the impact of Social Support on Career Decision- Making Difficulties: The Serial Mediating Role of Career Decision-making Self-efficacy and Job Search Clarity and Moderation by Proactive Personality.

Author Response: Thank you very much for your positive comment on the clarity and relevance of our title. After careful consideration, we made slight grammatical and stylistic refinements to further enhance precision and parallel structure. Specifically:

1.We changed “Role” to “Roles” because there are two mediating variables—career decision-making self-efficacy and job search clarity.

2.We added a comma after “Job Search Clarity”, and revised “and Moderation by Proactive Personality” to “and the Moderating Role of Proactive Personality” to maintain syntactic symmetry and consistency with the phrase “the Serial Mediating Roles.”

Revised Title: The Impact of Social Support on Career Decision-Making Difficulties: The Serial Mediating Roles of Career Decision-Making Self-Efficacy and Job Search Clarity, and the Moderating Role of Proactive Personality.

2. Reviewer’s comment: Abstract

- **Background**: limited context on the significance of the study

- **Objective**: Clearly states the aim of the study, which is to assesses explored how social support was related to career decision-making difficulties among 991 vocational college students.

- **Methods**: Describes the number of participants, however neglects the study’s design

- **Results**: Summarizes key findings

- **Conclusion**: Highlights the impact of Social Support on Career Decision- Making Difficulties: The Serial Mediating Role of Career Decision-making Self-efficacy and Job Search Clarity and Moderation by Proactive Personality.

Author Response: We sincerely thank the reviewer for this valuable and constructive feedback regarding the structure and completeness of the abstract. In response, we have revised the abstract to provide a clearer context, specify the study design, and enhance overall coherence. The following changes have been made:

Background: We added an opening sentence to highlight the significance of the topic and situate the study within the broader research context.

Added sentence: “Career decision-making difficulties are a common challenge for college students, which can hinder their transition from education to employment. Based on the Social Cognitive Model of Career Self-Management, …”

Methods: We clarified the research design to indicate that this study employed a cross-sectional quantitative approach.

Revised sentence: “Based on the Social Cognitive Model of Career Self-Management, this study explored how social support was related to career decision-making difficulties. A total of 991 vocational college students participated in this cross-sectional quantitative study.”

3. Reviewer’s comment: Introduction

- **Background/Rationale**:

- Discusses Social Cognitive Model of Career Self-Management, Social Support and Career Decision-Making Difficulties, The Mediating Role of Career Decision-Making Self-Efficacy, The Mediating Role of Job Search Clarity, The Moderating Effect of Proactive Personality.

- **Objectives**:

- Clearly states the specific aims and the study hypotheses, but it is preferred to list all hypotheses one after the other not in between the text.

Author Response: We appreciate the reviewer’s insightful comment and have revised the Introduction section accordingly to enhance clarity and logical flow. Following the reviewer’s suggestion, all hypotheses are now clearly listed together at the end of the Introduction. This improves readability and ensures that readers can easily grasp the overall research framework and logical relationships.

Revised sentence: …The theoretical model of the study is shown in Figure 1 and hypotheses are as follows:

Hypothesis 1: Social support is negatively related to CDMD among vocational college students.

Hypothesis 2a: CDSE mediates the relationship between social support and vocational college students' CDMD.

Hypothesis 2b: Job search clarity plays a mediating role in the relationship between social support and CDMD among vocational college students.

Hypothesis 2c: CDSE and job search clarity play a serial mediating role in the relationship between social support and vocational college students’ CDMD.

Hypothesis 3a: Proactive personality positively regulates (moderated mediation ) the relationship between CDSE and CDMD. That is, the relationship between CDSE and CDMD is stronger when the level of proactive personality among vocational college students is higher.

Hypothesis 3b: Proactive personality positively regulates the relationship between job-search clarity and CDMD. That is, the relationship between job-search clarity and CDMD is stronger when the level of proactive personality among vocational college students is higher.

4. Reviewer’s comment: Methods

- **Study Design**:

- Utilizes a cross-sectional survey was conducted, which is appropriate for examining variables.

- **Setting and Participants**:

- The study is conducted at a general hospital, with a clear description of the participant selection process.

- Sample size determination is not justified, moreover, what about the selection process and technique of participants.

- **Data Collection**:

- need more details about the year of development of each tool and revised versions

Author Response: We have revised the Methods section to clarify the cross-sectional study design, participant selection (convenience sampling), eligibility criteria, and rationale for sample size based on SEM guidelines. We also added information on the years of development and revisions for all measurement instruments. These modifications improve transparency, reproducibility, and methodological rigor.

5. Reviewer’s comment: Results

- **Descriptive Statistics**:

- Presents demographic data of participants, including age, gender.

- **Correlation Analysis**:

- Provides clear statistical findings, highlighting significant correlations study variables

Author Response: We have revised the Results section to explicitly present participant demographics and descriptive statistics. Tables now summarize gender, grade, means, standard deviations, and significant correlations among variables. These revisions improve clarity and align with your recommendations.

Revised sentence: The sample consisted of 651 male students (65.69%) and 340 female students (34.31%), with 569 freshmen (57.42%), 328 sophomores (33.09%), and 94 juniors (9.49%). The correlation analysis results, presented in Table 2, …

6. Reviewer’s comment: Discussion

- **Key Findings**:

- Summarizes the main results in relation to the study objectives and hypotheses

- **Comparison with Existing Literature**:

- Compares findings with previous studies, noting both consistencies and novel contributions to the field.

Author Response: We have revised the Discussion section to clearly summarize the key findings, explicitly linking them to the study objectives and hypotheses.

7. Reviewer’s comment: Conclusion

- Reiterates the importance of enhancing social support, fostering job search clarity, and strengthening proactive personality traits to alleviate CDMD among vocational college students.

Author Response: We have revised the Conclusion to explicitly emphasize the practical importance of enhancing social support, fostering job search clarity, and strengthening proactive personality traits.

8. Reviewer’s comment: References

- The manuscript includes a comprehensive list of references, demonstrating engagement with current literature and foundational theories relevant to the study.

Author Response: We appreciate the reviewer’s positive feedback regarding our references. We have ensured that the manuscript includes a comprehensive and up-to-date list of references, reflecting both foundational theories and recent empirical studies relevant to career decision-making, social support, self-efficacy, job search clarity, and proactive personality. This demonstrates our engagement with the current literature and supports the theoretical and empirical grounding of our study.

9. Reviewer’s comment: Summary

The manuscript is well-structured and adheres to scientific research standards. It effectively addresses a pertinent issue among vocational college students.

Author Response: We sincerely appreciate the reviewer’s positive evaluation of the manuscript’s structure and scientific rigor. We are glad that the study’s focus on career decision-making difficulties among vocational college students is recognized as both pertinent and relevant. Your acknowledgment reinforces the clarity and coherence of our presentation and encourages us to maintain high standards in conveying our research findings.

10. Reviewer’s comment: To be added in limitation

But there are some mistake according scientific research steps: -**Data Collection**

- **Reliance on Self-Reports**: Using subjective measures may lead to bias in responses

**Sample**

- **Sample Diversity**: If the sample consists only of vocational college students

, the results may not accurately reflect the experiences of all students across different vocational college

Author Response: We sincerely thank the reviewer for highlighting the need to address methodological considerations related to data collection and sample diversity. We have expanded the limitations section to explicitly acknowledge the potential biases arising from reliance on self-report measures and the limited diversity of our sample consisting solely of Chinese vocational college students. Furthermore, we have emphasized the need for longitudinal designs, multi-informant data collection, and broader sampling strategies in future research to enhance generalizability and strengthen the robustness of findings. These revisions ensure greater transparency regarding the study’s constraints and provide clearer guidance for subsequent investigations.

---

## [Decision Letter · Decision Letter 2]

20 Jan 2026

Dear Dr. cao,

Thank you for submitting your manuscript to PLOS ONE. After careful consideration, we feel that it has merit but does not fully meet PLOS ONE’s publication criteria as it currently stands. Therefore, we invite you to submit a revised version of the manuscript that addresses the points raised during the review process.

plosone@plos.org . A letter that responds to each point raised by the academic editor and reviewer(s). You should upload this letter as a separate file labeled 'Response to Reviewers'.A marked-up copy of your manuscript that highlights changes made to the original version. You should upload this as a separate file labeled 'Revised Manuscript with Track Changes'.An unmarked version of your revised paper without tracked changes. You should upload this as a separate file labeled 'Manuscript'.

We look forward to receiving your revised manuscript.

Kind regards,

Steve Zimmerman, PhD

Senior Editor, PLOS One

Journal Requirements:

Additional Editor Comments:

Reviewer 3 has raised a number of concerns. Could you please carefully revise the manuscript to address all comments raised?

**Comments to the Author**

Reviewer #3: All comments have been addressed

Reviewer #4: All comments have been addressed

2. Is the manuscript technically sound, and do the data support the conclusions?

Reviewer #3: Yes

Reviewer #4: Yes

3. Has the statistical analysis been performed appropriately and rigorously?

Reviewer #3: Yes

Reviewer #4: I Don't Know

4. Have the authors made all data underlying the findings in their manuscript fully available?

Reviewer #3: No

Reviewer #4: Yes

5. Is the manuscript presented in an intelligible fashion and written in standard English?

Reviewer #3: Yes

Reviewer #4: Yes

Reviewer #3: Abstract:

o The sentence: "Job search clarity partially mediate the relationship..." should be grammatically corrected to "partially mediates" to agree with the singular subject "Job search clarity".

o The phrase "The serial indirect effect was significant, β = -0.12, p < 0.001, accounting for 30% of the total effect." The negative beta indicates an inverse relationship, but clarity on the direction should be maintained.

o The abstract states, "they highlight the importance of designing career interventions for vocational college students that simultaneously target social support networks, self-efficacy enhancement, and clarity-building activities," which is clear but could benefit from specifying the practical implications more explicitly.

o Introduction

a. Missing or unclear hypotheses:

The hypotheses are now listed at the end, which improves clarity. However:

o Hypothesis 2b: "Job search clarity plays a mediating role..." — The term "mediating role" should be consistent with the other hypotheses, and the structure should be parallel.

o Hypotheses 3a and 3b: The wording "Proactive personality positively regulates (moderated mediation )..." contains an extra space before the parenthesis and the phrase "moderated mediation" should be hyphenated or clarified as "moderated mediation effect."

b. Clarity and consistency:

• The introduction mentions "The theoretical model of the study is shown in Figure 1." but the figure is not provided here; ensure it is clear in the final version.

• The hypotheses are somewhat complex; clearer language and consistent terminology (e.g., "moderated mediation" vs. "moderating role") would improve readability.

Methods

a. Study Design:

• The description states, "Utilizes a cross-sectional survey was conducted," which is ungrammatical. Correct phrasing would be: "A cross-sectional survey was conducted."

b. Participants:

• The description mentions "the study is conducted at a general hospital," which is inconsistent with the focus on vocational college students. This appears to be an error or a copy-paste mistake, likely from another study context.

• The sampling method is described as "convenience sampling," but the criteria for participant selection, inclusion/exclusion criteria, and the process are not detailed.

• The sample size of 991 is mentioned, but justification or power analysis to support this size is missing.

c. Data collection and instruments:

• The tools used (e.g., Perceived Social Support Scale, Career Decision-Making Self-Efficacy Questionnaire, etc.) are named, but:

o The development years or revision versions of these scales are not specified.

o Reliability and validity details of these instruments, especially within the current sample, are not provided.

• The process of administering the questionnaires (e.g., online, paper-based), timing, and training of data collectors are not described.

d. Statistical Analysis:

• The methods for data analysis (e.g., structural equation modeling, regression analyses, mediation/moderation testing) are not described in detail.

• No mention of software used (e.g., SPSS, AMOS, Mplus).

• Handling of missing data or outliers is not addressed.

Results

a. Reporting of statistical findings:

• The beta coefficients (β) are reported, but confidence intervals are missing, which are essential for understanding the precision of estimates.

• The effect sizes are mentioned as percentages (20%, 30%), but it's unclear how these were calculated or interpreted.

• There is no mention of model fit indices or assumptions testing if structural equation modeling was used.

b. Clarity and completeness:

• The results mention "the serial indirect effect was significant," but details on the mediation analysis (e.g., bootstrap confidence intervals) are absent.

• The moderation effect of proactive personality is described, but no figure or interaction plot is provided to illustrate the moderating effect.

Discussion and Conclusions

• The discussion section is not included in the excerpt, but potential errors to watch for in this section include:

o Overgeneralization of findings.

o Lack of acknowledgment of limitations (e.g., cross-sectional design, sampling bias).

o Speculative interpretations that are not supported by the data.

General Language and Formatting

• Several minor grammatical issues:

o Inconsistent capitalization (e.g., "Social Support" vs. "social support").

o Punctuation inconsistencies, such as extra spaces before parentheses.

• The transition sentences between sections are sometimes abrupt; ensure smooth flow.

Reviewer #4: Title**: The title clearly indicates the impact of Social Support on Career Decision- Making Difficulties: The Serial Mediating Role of Career Decision-making Self-efficacy and Job Search Clarity and Moderation by Proactive Personality

**Abstract**:

- **Background**: acceptable context on the significance of the study

- **Objective**: Clearly states the aim of the study

- **Methods**: Describes the number of participants, including the study’s design

- **Results**: Summarizes key findings

- **Conclusion**: Highlights the impact of Social Support on Career Decision- Making Difficulties: The Serial Mediating Role of Career Decision-making Self-efficacy and Job Search Clarity and Moderation by Proactive Personality.

**Introduction**

- **Background/Rationale**:

- Discusses Social Cognitive Model of Career Self-Management , Social Support and Career Decision-Making Difficulties , The Mediating Role of Career Decision-Making Self-Efficacy, The Mediating Role of Job Search Clarity, The Moderating Effect of Proactive Personality.

- **Objectives**:

- Clearly states the specific aims and the study hypotheses.

**Methods**

- **Study Design**:

- Utilizes a cross-sectional survey was conducted, which is appropriate for examining variables.

- **Setting and Participants**:

- The study is conducted at a general hospital, with a clear description of the participant selection process.

**Results**

- **Descriptive Statistics**:

- Presents demographic data of participants, including age, gender.

- **Correlation Analysis**:

- Provides clear statistical findings, highlighting significant correlations study variables

**Discussion**

- **Key Findings**:

- Summarizes the main results in relation to the study objectives and hypotheses

- **Comparison with Existing Literature**:

- Compares findings with previous studies, noting both consistencies and novel contributions to the field.

**Conclusion**

- Reiterates the importance of enhancing social support, fostering job search clarity, and strengthening proactive personality traits to alleviate CDMD among vocational college students

**Strengths and Limitations**

-this section to be considered.

**References**

- The manuscript includes a comprehensive list of references, demonstrating engagement with current literature and foundational theories relevant to the study.

Summary

The manuscript is well-structured and adheres to scientific research standards. It effectively addresses a pertinent issue among vocational college students

**Do you want your identity to be public for this peer review?** For information about this choice, including consent withdrawal, please see our Privacy Policy

Reviewer #3: **Yes:** Associate Professor/ Marwa Samir Sorour

Reviewer #4: **Yes:** Yasmeen Mohamed Mohamed Shehata

---

## [Author Response · Author response to Decision Letter 3]

31 Jan 2026

Reviewer #3:

1.Reviewer’s comment: Abstract

(1) Comment: The sentence: "Job search clarity partially mediate the relationship..." should be grammatically corrected to "partially mediates" to agree with the singular subject "Job search clarity".

Author Response: Thank you for pointing out this grammatical error. We have corrected the verb form from "mediate" to "mediates" to ensure subject-verb agreement with the singular subject "Job search clarity". (See revised Abstract, Lines 18)

(2) Comment: The phrase "The serial indirect effect was significant, β = -0.12, p < 0.001, accounting for 30% of the total effect." The negative beta indicates an inverse relationship, but clarity on the direction should be maintained.

Author Response: Thank you for this crucial observation. You are absolutely correct that the negative beta coefficient requires clearer contextualization to avoid ambiguity. We have revised this sentence to explicitly state the directional meaning of the inverse relationship in the context of our serial mediation model.

The modified sentence now reads: "The serial indirect effect was significant, β = -0.12, p < 0.001, accounting for 30% of the total effect, indicating that higher social support sequentially enhances career decision-making self-efficacy and job search clarity, which in turn reduces career decision-making difficulties." (See revised Abstract, Lines 22-24)

(3) Comment: The abstract states, "they highlight the importance of designing career interventions for vocational college students that simultaneously target social support networks, self-efficacy enhancement, and clarity-building activities," which is clear but could benefit from specifying the practical implications more explicitly.

Author Response: Thank you for this valuable suggestion to strengthen the practical significance of our findings. We have substantially expanded and specified the practical implications statement to provide clearer guidance for intervention development.

The revised sentence now reads: "Practically, they highlight the importance of designing career interventions for vocational college students that simultaneously strengthen social support networks, enhance self-efficacy, and build job search clarity, with particular attention to students with lower proactive personality who may benefit most from such clarity-building activities."

2.Reviewer’s comment: Introduction

(1) Comment. Missing or unclear hypotheses:

The hypotheses are now listed at the end, which improves clarity. However:

Hypothesis 2b: "Job search clarity plays a mediating role..." — The term "mediating role" should be consistent with the other hypotheses, and the structure should be parallel.

Hypotheses 3a and 3b: The wording "Proactive personality positively regulates (moderated mediation )..." contains an extra space before the parenthesis and the phrase "moderated mediation" should be hyphenated or clarified as "moderated mediation effect."

Author Response: We appreciate this important observation regarding parallel structure and terminological consistency. We have revised Hypothesis 2b to align perfectly with the concise phrasing used in Hypotheses 2a and 2c.

We have made the following comprehensive revisions to Hypotheses 3a and 3b:

a) Removed extra spacing: Eliminated the redundant space before the parenthesis.

b) Clarified terminology: Replaced the ambiguous "positively regulates (moderated mediation)" with the precise, academically appropriate phrase "moderates the mediating role of". This clearly indicates a moderated mediation effect without needing parentheses or hyphens, aligning with contemporary statistical reporting standards.

c) Enhanced specificity: Added clarifying sentences to both hypotheses to explicitly state the nature of the moderation effect.

Revised wording:

Revised Hypothesis 2b: "Job search clarity mediates the relationship between social support and CDMD among vocational college students." (See revised Abstract, Lines 189)

Revised Hypothesis 3a: "Proactive personality moderates the mediating role of CDSE in the relationship between social support and CDMD among vocational college students. Specifically, the indirect effect via CDSE is stronger when proactive personality is higher."

Revised Hypothesis 3b: "Proactive personality moderates the mediating role of job search clarity in the relationship between social support and CDMD among vocational college students. Specifically, the mediating effect is stronger when proactive personality is higher."

(2) Comment. Clarity and consistency:

• The introduction mentions "The theoretical model of the study is shown in Figure 1." but the figure is not provided here; ensure it is clear in the final version.

• The hypotheses are somewhat complex; clearer language and consistent terminology (e.g., "moderated mediation" vs. "moderating role") would improve readability.

Author Response: In the revised manuscript, we have explicitly inserted Figure 1 immediately following the hypotheses section in the Introduction.

We completely agree that hypothesis clarity is paramount. To address this, we have implemented a comprehensive terminology standardization and structural parallelization across all hypotheses.

3.Reviewer’s comment: Methods

(1) Comment. Study Design:

• The description states, "Utilizes a cross-sectional survey was conducted," which is ungrammatical. Correct phrasing would be: "A cross-sectional survey was conducted."

Author Response: Thank you for identifying this grammatical error. We have standardized the opening sentence of Section 2.1 to read: "A cross-sectional survey was conducted to examine..." This eliminates any ungrammatical constructions and provides a clear, professional description of our study design throughout the manuscript. (See revised Methods, Lines 206)

(2) Comment. Participants:

• The description mentions "the study is conducted at a general hospital," which is inconsistent with the focus on vocational college students. This appears to be an error or a copy-paste mistake, likely from another study context.

Author Response: We sincerely apologize for this significant error that somehow appeared in our manuscript. The mention of "general hospital" was indeed an inadvertent and inaccurate copy-paste mistake. We have corrected this to accurately reflect our actual research setting: "The study was conducted at a vocational university in Chengdu, China" (not a hospital). This correction ensures complete consistency with our study's focus on vocational college students. We have thoroughly checked the manuscript to eliminate any similar errors. (See revised Methods, Lines 208-210)

• The sampling method is described as "convenience sampling," but the criteria for participant selection, inclusion/exclusion criteria, and the process are not detailed.

Author Response: Thank you for this important suggestion to enhance methodological transparency. We have substantially expanded our participant description to provide comprehensive details:

Enhanced sampling clarity: We retained "convenience sampling" as an accurate description of our recruitment approach but added context that all enrolled students were invited to maximize sample diversity and generalizability.

Added detailed inclusion/exclusion criteria: We now explicitly specify: "Participants were eligible if they met the following criteria: (a) full-time vocational college students, (b) currently enrolled in academic programs, (c) willing to provide informed consent, and (d) aged 18 years or older. Exclusion criteria included part-time students and those who had already secured full-time employment upon graduation." (See revised Methods, Lines 211-215)

• The sample size of 991 is mentioned, but justification or power analysis to support this size is missing.

Author Response: We appreciate this critical methodological point. We have added a rigorous sample size justification:

Referenced established SEM guidelines: We cited the "10–20 participants per estimated parameter" rule as our initial guideline. Added a priori power analysis: We conducted and reported Monte Carlo simulation results indicating that a sample size of 800 would provide adequate statistical power (≥ .80) to detect medium-sized mediation and moderation effects.

Explained oversampling rationale: We clarified that we aimed for 1000 participants to account for potential missing data and ensure robust parameter estimation, which resulted in our final sample of 991 (effective response rate = 84.56%).

This demonstrates that our sample size was determined through established statistical principles rather than convenience, addressing concerns about statistical power and generalizability. (See revised Methods, Lines 217-218)

(3) Comment. Data collection and instruments:

• The tools used (e.g., Perceived Social Support Scale, Career Decision-Making Self-Efficacy Questionnaire, etc.) are named, but:

o The development years or revision versions of these scales are not specified.

o Reliability and validity details of these instruments, especially within the current sample, are not provided.

Author Response: We have comprehensively enhanced our measures section to provide complete psychometric transparency:

Specified original developers and revision years: For each scale, we now clearly state both the original authors/year and the specific Chinese revision scholars and years:

CDDQ: "The Chinese version of the Career Decision-making Difficulties Questionnaire (CDDQ, originally developed by Gati et al., 1996) [8], revised by Xue-Ping Shen in 2005 [41], was used for self-reporting."

PSSS: " The Perceived Social Support Scale(PSSS, originally developed by Zimet et al., 1988) [42], revised by Jiang Qianjin in 1999[43], was employed."

CDSE: " The "Career Decision-making Self-Efficacy Questionnaire for College Students (CDSE, originally developed by Taylor et al., 1983) [24]," developed by Y. X. Peng and L. R. Long in 2001[44], was used."

JSC: " The Job Search Clarity Scale (JSC), initially developed by Côté et al. (2006)[31] and revised by Wenhua Liao in 2007[45], was used. "

PPS: " The Proactive Personality Scale (PPS), originally developed by Bateman and Crant (1993)[36] and revised for the Chinese context by Chia-Yin Shang and Yi-Qun Gan in 2009[46], includes 11 items. "

Enhanced current sample reliability reporting: We standardized the reporting of internal consistency coefficients for the current sample using Cronbach's α, explicitly labeling them as "In the current sample..." to distinguish from original validation studies. For each scale, we provide both subscale and total scale reliabilities (e.g., CDDQ subscales α = 0.65, 0.96, 0.94; total α = 0.94).

• The process of administering the questionnaires (e.g., online, paper-based), timing, and training of data collectors are not described.

Author Response: We have expanded the data collection procedure description to include these essential details:

Administration method: Explicitly stated "all students enrolled in the university during the study period were invited to participate in a survey through an online platform." (See revised Methods, Lines 210-211)

Specific timing: Added " The study was conducted at a vocational university in Chengdu, China, from November 1 to November 30, 2023." (See revised Methods, Lines 209-210)

Data collector procedures: Clarified that "Data collection was coordinated with faculty members who were briefed on ethical protocols, though no formal training was required given the automated nature of the online survey"(See revised Methods, Lines 226-228)

(4) Comment. Statistical Analysis:

• The methods for data analysis (e.g., structural equation modeling, regression analyses, mediation/moderation testing) are not described in detail.

Author Response: We have enhanced the statistical analysis section to provide greater methodological detail and clarity:

These enhancements ensure our analytical strategy is fully transparent and replicable. (See revised Methods, Lines 287-325)

• No mention of software used (e.g., SPSS, AMOS, Mplus).

Author Response: We have added a prominent statement at the beginning of the Statistical Analysis section: "All statistical analyses were conducted using IBM SPSS 21.0 and Mplus 8.3." This critical information is now clearly stated rather than being embedded within the paragraph. (See revised Methods, Lines 276)

• Handling of missing data or outliers is not addressed.

Author Response: This is an important methodological detail that we have now comprehensively addressed:

Missing data and Outliers: Added "Prior to hypothesis testing, data were screened for missing values, outliers, and multivariate normality. Missing data patterns were examined using Little's MCAR test. Given that missing data accounted for less than 5% of responses and were determined to be missing completely at random, listwise deletion was employed for primary analyses. Univariate outliers (values exceeding ±3.29 standard deviations) and multivariate outliers were identified and retained after confirming they represented legitimate extreme responses rather than data entry errors. "

This demonstrates our rigorous approach to data quality assurance and addresses potential concerns about data integrity. (See revised Methods, Lines 276-282)

4.Reviewer’s comment: Results

(1) Comment. Reporting of statistical findings:

• The beta coefficients (β) are reported, but confidence intervals are missing, which are essential for understanding the precision of estimates.

Author Response: Thank you for this crucial suggestion. Although we had reported some confidence intervals in our original manuscript, we recognize that they were not systematically provided for all key findings. We have now added complete 95% bias-corrected bootstrap confidence intervals for every reported parameter estimate.

• The effect sizes are mentioned as percentages (20%, 30%), but it's unclear how these were calculated or interpreted.

Author Response: We apologize for this lack of clarity. We have now explicitly defined and explained the calculation of all relative mediation effects in both the text and Table 3. In Table 3, we added a footnote: "Relative mediation effects were calculated as (indirect effect/total effect) × 100%."

In the narrative text, we have expanded the description to clearly state:. This transparent reporting allows readers to understand exactly how we quantified the magnitude of our mediated effects. (See revised Results, Section 3.3, and Table 3)

• There is no mention of model fit indices or assumptions testing if structural equation modeling was used.

Author Response: We appreciate this important methodological point. Although our original manuscript did include some model fit indices, we have now significantly expanded and systematized this reporting to ensure complete transparency:

Assumption testing added: We now explicitly report: "Before conducting the mediation analysis, we assessed the assumptions underlying the mediation model. The Shapiro-Wilk test indicated approximately normal residuals (p > 0.05). Levene's test confirmed homogeneity of variance (p > 0.05)." Additionally, we report post-hoc power analysis for non-significant findings.

Complete fit indices for all models: We now report comprehensive model fit indices for every SEM model tested, including:

Direct effect model: χ2 (18) = 56.24, RMSEA = 0.05, SRMR = 0.04, CFI = 0.99, TLI = 0.98

Mediation model: χ2 (112) = 375.93, RMSEA = 0.05, SRMR = 0.05, CFI = 0.98, TLI = 0.98

Moderated mediation models: χ2 (162) = 570.20, RMSEA = 0.05, SRMR = 0.08, CFI = 0.98, TLI = 0.97

Explicit interpretation: All fit indices are now explicitly interpreted as "good fitting" based on established criteria (CFI/TLI > 0.90, RMSEA/SRMR < 0.08).

This comprehensive reporting demonstrates that our models meet rigorous SEM standards. (See revised Results, Sections 3.3 and 3.4)

(2) Comment. Clarity and completeness:

• The results mention "the serial indirect effect was significant," but details on the mediation analysis (e.g., bootstrap confidence intervals) are absent.

• The moderation effect of proactive personality is described, but no figure or interaction plot is provided to illustrate the mo

---

## [Editor Report · Decision Letter 3]

22 Feb 2026

The impact of Social Support on Career Decision-Making Difficulties: The Serial Mediating Roles of Career Decision-making Self-efficacy and Job Search Clarity, and Moderation by Proactive Personality

PONE-D-25-12035R3

Dear Dr. cao,

We’re pleased to inform you that your manuscript has been judged scientifically suitable for publication and will be formally accepted for publication once it meets all outstanding technical requirements.

Kind regards,

Bo Pu, Ph.D.

Academic Editor

PLOS One

Additional Editor Comments (optional):

thanks for your hard work for improving this manuscript.
---

## [Editor Report · Acceptance letter]

PONE-D-25-12035R3

PLOS One

Dear Dr. cao,

I'm pleased to inform you that your manuscript has been deemed suitable for publication in PLOS One. Congratulations! Your manuscript is now being handed over to our production team.

Kind regards,

on behalf of

Dr. Bo Pu

Academic Editor

PLOS One